# Investigation of the Efficiency of Satellite-Derived LST Data for Mapping the Meteorological Parameters in Istanbul

**Adalet Dervisoglu**

Geomatics Engineering Department, Civil Engineering Faculty, Istanbul Technical University, Istanbul 34467, Turkey; adervisoglu@itu.edu.tr

**Abstract:** Land surface temperature (LST) is an essential parameter for studying environmental and ecological processes and climate change at various scales. It is also valuable for studies of evapotranspiration, soil moisture conditions, surface energy balance, and urban heat islands. Since meteorological station data can provide a limited number of point data, satellite images that provide high temporal and spatial resolution LST data in large areas are needed to be used in all these applications. In this study, the usage of satellite-derived LST images was investigated in comparison with meteorological station data measurements in Istanbul, which has heterogeneous urban structures. LST data were obtained from Landsat 5 TM, Landsat 8 OLI/TIRS, and Terra Moderate Resolution Imaging Spectroradiometer (MODIS) satellite images using the Google Earth Engine (GEE) cloud platform. The linear correlation analysis performed between Landsat LST and MODIS LST images gave a high correlation (r = 0.88). In the correlation analysis, hourly air temperature and soil temperature meteorology station data provided by the State Meteorological Service and LST values obtained from images taken from Landsat TM/TIRS and Terra MODIS were used. The correlations between air temperatures and Landsat LST ranged from 0.47–0.95 for 1987–2017 to 0.44–0.80 for MODIS LST for 2000–2017. The correlations between 5 cm soil temperatures and Landsat LST ranged from 0.76–0.93 for 2009–2017 to 0.22–0.61 for MODIS LST 2000–2017. In addition, linear regression models produced with meteorological parameters and LST values were applied to 2022 LST maps to show the spatial distribution of these parameters, and then, accuracy analyses were made.

**Keywords:** LST; TM/TIRS; MODIS; linear regression; Istanbul; Turkey; air temperature; soil temperature

## 1. Introduction

Land surface temperature (LST) is a measure of the temperature of the Earth's surface. It is an important variable in meteorology and climate science as it affects the energy and water exchange between the atmosphere and the Earth's surface. Accurately understanding LST at the global and regional levels helps evaluate land surface–atmosphere change processes in models and, when combined with other physical properties, such as vegetation and soil moisture, provides a valuable measure of surface conditions [1]. LST is typically measured using satellite imagery or ground-based sensors. While evaporation and urban heat islands can be monitored with LST, it is also used in agriculture, disaster response, and urban planning applications. Remote sensing is exceptionally beneficial for understanding spatiotemporal land cover change in relation to key physical properties in terms of surface brightness and emission data. Optical remote sensing images provide the reflectivity of the Earth's surface, while thermal infrared images show the emissivity of the surface material, and both complement each other and are essential for environmental monitoring [2]. Satellite-derived surface temperature products are widely used for different purposes in determining both land and sea surface temperature, with the advantage of providing continuous monitoring of satellites over large regions: in plant or agricultural studies [3–6], climate change studies [7,8], sea surface temperature determination studies [9–12], urban studies [13–16], forest and forest fires [17,18], analysis of annual land cover dynamics [19,20], and drought analysis [21,22].

Obtaining ground temperature data at meteorological stations is time-consuming and costly and cannot reflect the surface temperature on a large scale due to the heterogeneity of the Earth (vegetation, topography, and soil moisture) [23]. However, due to the spatial resolution of satellite images (each pixel of the image covers an area of several hundred meters or kilometers), point temperature measurement at the weather station may not coincide with the temperature of a pixel.

Because LSTs from satellite data contain various corrections, their accuracy must be validated in order to provide reliable information about the quality of the LST for the above-mentioned applications. Validation is the process of independently assessing the uncertainty of data from system outputs, and without validation, any data derived from remotely sensed data cannot be used safely. For this reason, various methods are used to verify the LST values derived from the satellite; the temperature-based method (T-based), radiation-based method (R-based), and cross-validation are commonly used methods [24]. The T-based, also used in this study, is a ground-based method that directly compares the satellite-derived LST with a sufficient number of in situ LST measurements at the satellite overpass [25]. Many researchers have studied the relationships between LST derived from satellites, such as MODIS and Landsat satellites, and near-surface air temperature obtained from ground-based meteorological stations. These strong relationships were demonstrated by statistical and regression analysis. In a study comparing the LST values derived from MODIS and ground-based near-surface air measurements obtained from 14 observation stations covering coastal, mountainous, and urban areas of Cyprus, it was reported that there is a very high correlation between the data [26]. In a study covering the whole of Portugal, a 10-year forecast was made using a statistical approach based on the correlation of MODIS LST data with data from 106 meteorological stations (i.e., minimum, maximum, and average air temperatures between 2000 and 2009). As a result of the analysis, they showed the strong relationship between the MODIS LST and air temperature (as an example, for the average temperature, 85% of the stations had $r^2$ greater than 0.90 and RMSE less than 1.5 °C) and validated the accuracy of the LST values [27]. In a study conducted in the Eastern Thrace region, it was found that 27 meteorological stations' data (monthly average temperature, precipitation, and relative humidity) and Landsat LST data showed a positive correlation between satellite-based LST and ground temperature values and a negative correlation for precipitation and relative humidity [28]. In a study, the LSTs of Antalya central districts were examined using Landsat 7, Landsat 8, and MODIS satellite images, and a high correlation was observed between MODIS and Landsat 7 ($r^2$ = 0.7) and MODIS and Landsat 8 ($r^2$ = 0.9) [16]. In a study conducted in China, Landsat 8 data and four different methods were used to derive LST values. Temperatures from local weather stations of the China Meteorological Data Network were used for validation, and MODIS daily LSTs were used for cross-validation. Comparing Landsat LST values with air temperature data and MODIS data, correlations ranging from 0.82 to 0.85 were obtained [29]. In another study conducted to estimate LST from Landsat 8 data in arid lands, soil temperatures of three meteorological stations at a depth of 5 cm were used for temperature-based verification, and statistical coefficients were found to be higher than 0.87 in all methods [23].

It is significant to reveal the relationship between urban areas and LST, especially in complex urban metropolitan areas with heterogeneous surface classes where rapid urbanization occurs. However, in cases with insufficient measurements from meteorological stations, statistical estimation of these parameters depending on previous years is also very important. By drawing attention to these two points, this study, which was carried out in Istanbul, a city with a very complex urban structure, aims to obtain meteorological parameter maps with acceptable accuracy from satellite data. Istanbul is a mega city with a population of 15.5 million and one of the fastest-growing cities in Europe. The study area lies between the continents of Europe and Asia and has an area of approximately 5757 km² [13]. With the rapidly increasing population and urbanization, the surface temperatures of Istanbul are also increasing due to the increasing number of buildings

and artificial surfaces. Therefore, this study is important in terms of obtaining accurate and reliable surface temperature data in complex urban environments such as Istanbul. In the study, together with the meteorological station data used for verification, LSTs were produced from Landsat and Terra MODIS satellite images (Figure 1) for the province of Istanbul, and the linear correlations between them were examined.

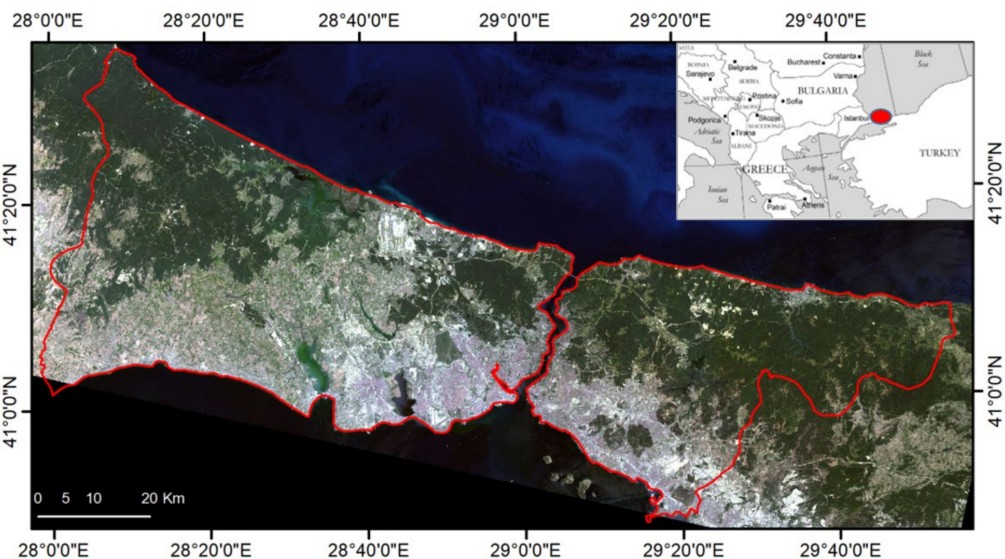

**Figure 1.** Study area on Landsat 8 OLI true color image (RGB:432) dated 23 July 2022, Istanbul.

This study has four main objectives: (i) to demonstrate the effectiveness of satellite-derived LST data; (ii) to examine the relationships between meteorological station data in Istanbul (i.e., air temperature and soil temperature) and satellite-derived LSTs; (iii) to produce 2022 maps for these two parameters using regression models applied to the 2022 LST maps (i.e., by performing regression models between meteorological station data and satellite-derived LSTs from previous years); and (iv) to validate the spatial distribution map of meteorological parameters with the actual meteorological station data.

## 2. Materials and Methods

### 2.1. Materials

In the study, Landsat 5 TM [30], Landsat 8 OLI-TIRS [31] thermal bands, and Terra Moderate Resolution Imaging Spectroradiometer (MODIS) MOD11A1 [32,33] V6.1 (daytime data) were used. The characteristics of the satellites used are given in Table 1.

The study was carried out on the Google Earth Engine (GEE) cloud platform. GEE is a cloud-based computing platform where various geospatial analyses can be performed on Google's cloud infrastructure. GEE combines a multi-petabyte catalog of satellite imagery and geospatial datasets with planetary-scale analysis capabilities [34]. GEE allows the filtering of low-quality images (such as shadow and cloud).

Using data in the clear pixel is essential for accurate surface temperature calculations using satellites; the accuracy of results will be reduced if the data have a pixel with clouds. The clearest sky of the year in the Istanbul region is in July; during this month, the sky is 95% clear, mostly clear, or partly cloudy [35], and for this reason, the month of July was chosen as the study period. The Landsat satellite images used in this study were determined by examining the cloudless July images between 1985 and 2022 in the study area individually in the USGS Earth Explorer [36], and the chosen images were also checked and used in GEE.

**Table 1.** Characteristics of Landsat 5 TM, Landsat 8 OLI/TIRS, and Terra MODIS MOD11A1.

| Satellite | Spectral Resolution (μm) | Spatial Resolution (m) | Temporal Resolution (Day) | Radiometric Resolution (Bit) |
|---|---|---|---|---|
| Landsat 5 TM | 6 Optical Bands (0.45–2.35) | 30 m | 16 | 8 |
| | 1 Thermal Band (10.40–12.50) | 120 m | | |
| Landsat 8 OLI/TIRS | 9 Optical Bands (0.43–2.30) | 30 m / 15 m | 16 | 16 |
| | 2 Thermal Bands (10.60–12.51) | 100 m | | |
| **Earth Science Data Type** | **Spectral Resolution (μm)** | **Spatial Resolution** | **Temporal Resolution** | **Number Type** |
| MOD11A1 | 2 Thermal Bands (31, 32) (10.780–12.270) | 1 km (actual, 0.928 km) | Daily (daytime) | Uint 16 (unsigned integer number) |

Meteorology station data (hourly air temperature, 5 cm soil temperature, relative humidity, evaporation) were requested from the Turkish General Directorate of Meteorology (TGDM) between 1987 and 2022. The stations whose data are used are given in Figure 2, with their locations and international meteorological station numbers [37].

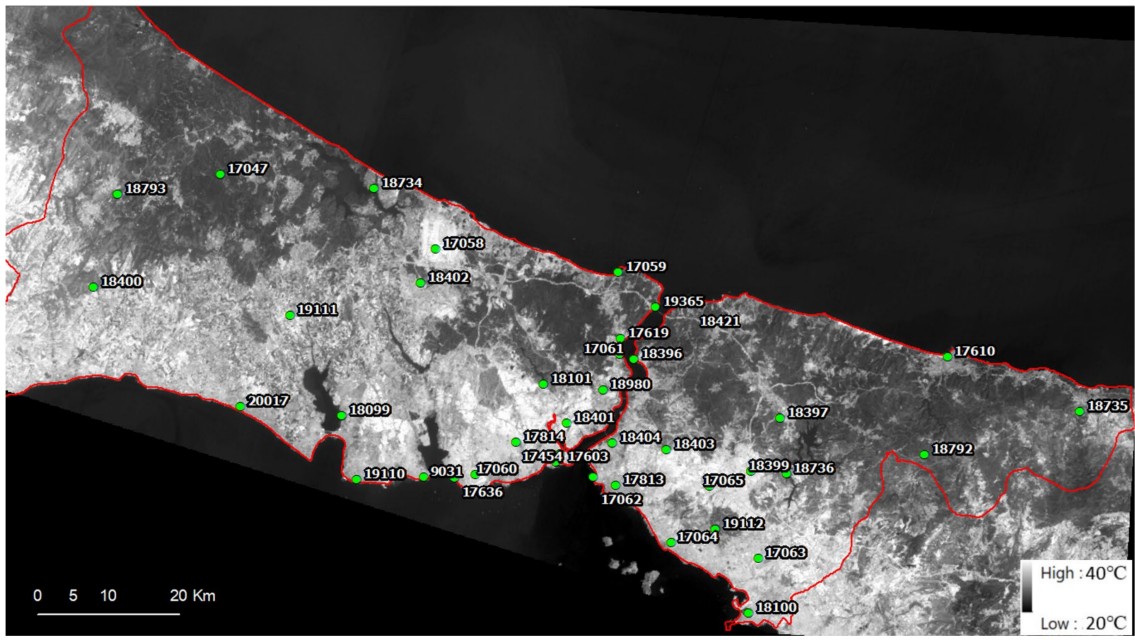

**Figure 2.** Meteorological stations with "International Meteorological Station Numbers" on 23 July 2022 Landsat LST image of Istanbul.

TGDM has fifty-four meteorological stations in Istanbul, and hourly meteorological parameters of all stations were requested for the study. However, it can be seen that hourly measurements were made at seven stations between 1987 and 2000 for the air temperature parameter, and the number of data increased with the establishment of automatic stations since 2003 (i.e., total numbers of stations are 2003: 11, 2009: 12, 2013: 18, 2017: 35, 2020: 41, 2022: 39). Since four stations are located on the breakwater, their data were not used in the analysis. In addition, it was observed that there were very high differences (up to

8–10 degrees) between the air temperatures measured at the meteorology stations at the airports and the LSTs, and these data were also not used in the analysis. Figure 2 shows the stations from which the data were gathered. Considering the hourly 5 cm soil temperature data, there are 7 stations for 1987, 8 stations for 2022, and 7 stations for other years on average. However, since there are few hourly relative humidity and evaporation data for very few dates in the study, these data could not be used. The air temperatures at the stations are measured at 2 m from the ground, and the meteorology stations are located at different altitudes (2 m to 381 m).

### 2.2. Methods

The hourly air temperature and hourly soil temperature (5 cm depth) data for the stations between 1987 and 2022 were obtained from the TGDM. Istanbul LST maps were generated in GEE for nine dates (23 July 1987; 7 July 1993; 18 July 1997; 26 July 2000; 3 July 2003; 19 July 2009; 30 July 2013; 9 July 2017; 23 July 2022) with Landsat 5 TM and Landsat 8 OLI/TIRS images and six dates (26 July 2000; 3 July 2003; 19 July 2009; 30 July 2013; 9 July 2017; 23 July 2022) with MODIS Terra images. Landsat 8 OLI/TIR image acquisition time for Istanbul is approximately 09.00 a.m. UTC, while Terra MODIS's is approximately 11:30 a.m. UTC (İstanbul local time: UTC+3 h). The Landsat frame used for Istanbul is Path:180, Row: 31, and image acquiring times for Landsat 5 TM are 8:07–8:34 a.m. UTC and Landsat 8 OLI/TIRS 08.45–08.47 a.m. UTC. Satellite image acquisition times were taken into account in the analyses. The flow chart of the study is given in Figure 3.

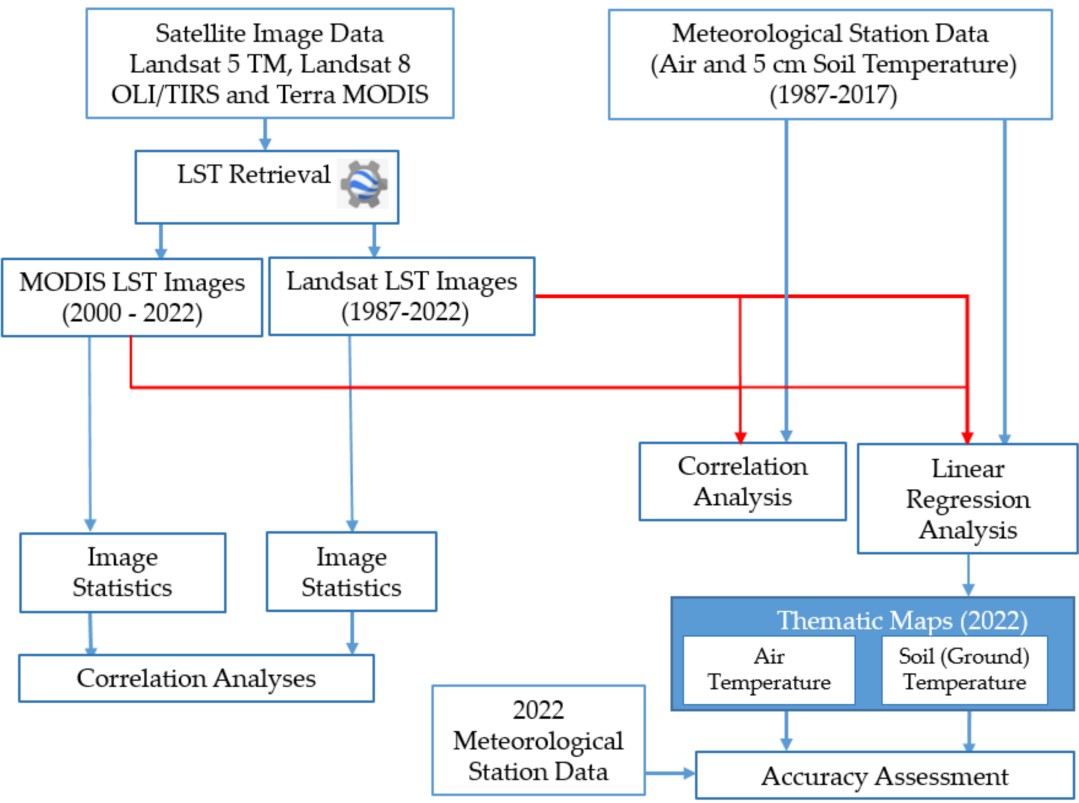

**Figure 3.** The flow chart of the study.

### 2.2.1. LST Retrieval

In the study, LSTs from Landsat thermal bands were calculated on the GEE platform, and LST images were generated. Mono-window algorithm [38], split-window algorithm [39,40], single-channel algorithm [41], radiative transfer equation [42], temperature-independent spectral indices method [43], and the inversion of Planck's function [44] are

the algorithms used for extracting land surface temperature (LST) from satellite imagery. In this study, the Planck function is utilized for calculating LST. According to Planck's radiation law, every object emits radiation at a certain temperature (not equal to 0 K) [45]. The process steps for calculating land surface temperatures are the conversion of digital values (DN) to spectral radiance values, the conversion of spectral radiance values to brightness temperature values, and the calculation of surface emissivity ($\varepsilon$) values [46]. For calculating LST, brightness temperature and emissivity values are required. Firstly, Landsat data is converted to TOA spectral radiance using the radiance rescaling factors with Equation (1) [47]:

$$L_\lambda = M_L \times Q_{cal} + A_L \tag{1}$$

where $L_{(\lambda)}$ is the top of atmosphere (TOA) spectral radiance (Watts/(m$^2$ * srad * μm), $M_L$ is the band-specific multiplicative rescaling factor ($M_L$ = 0.055375 for Landsat 5, $M_L$ = 0.0003342 for Landsat 8), $A_L$ is the band-specific additive rescaling factor ($A_L$ = 1.18243 for Landsat 5, $A_L$ = 0.1 for Landsat 8), $Q_{cal}$ is the quantized and calibrated standard product pixel values (DN). After that, thermal band data are converted to brightness temperature [47]:

$$\mathrm{T}_B = \frac{K_2}{\ln\left(\frac{K_1}{L_\lambda} + 1\right)} \tag{2}$$

where $\mathrm{T}_B$ is the brightness temperature in Kelvin, and $K_1$ and $K_2$ are the band-specific thermal conversion constants from the metadata file of USGS [48] ($K_1$ = 607.76, $K_2$ = 1260.56 for Landsat 5 TM, $K_1$ = 774.89, $K_2$ = 1321.08 for Landsat 8 OLI/TIRS).

In the literature, the normalized difference vegetation index (NDVI) thresholds method is most often used for estimating land surface emissivity ($\varepsilon$) [49]. When the NDVI is known of a surface area, the emissivity value can be assigned. In this study, land surface emissivity is calculated with NDVI and $P_V$ (vegetation proportion) values with thresholds and formulas given in Table 2.

$$NDVI = \frac{NIR - RED}{NIR + RED} \tag{3}$$

$P_V$ is calculated via the equation:

$$P_V = \left[\frac{NDVI - NDVI_{min}}{NDVI_{max} - NDVI_{min}}\right]^2 \tag{4}$$

$$LST = \frac{B}{1 + \left(\frac{\lambda \times T_B}{\rho}\right) \cdot \ln(\varepsilon)} \tag{5}$$

where $\lambda$ is the average wavelength of the band (μm);

$\rho$ = h * c/σ = $1.438 \times 10^{-2}$ m * K;

σ = Boltzmann constant ($1.38 \times 10^{-23}$ J/K);

h = Planck's constant ($6.626 \times 10^{-34}$ J * s);

c = the velocity of the light ($2.998 \times 10^8$ m/s).

**Table 2.** NDVI threshold model used in this study [11,48,50].

| NDVI Threshold | Land Cover Type | Surface Emissivity |
|---|---|---|
| NDVI < 0 | Water | 0.985 |
| $0 \leq NDVI \leq 0.1$ | Bare soil | ρR (red reflectance band) |
| $0.1 \leq NDVI \leq 0.7$ | Vegetation mixed with soil | $0.990P_V + 0.984(1 - P_V) + 0.04P_V(1 - P_V)$ |
| NDVI > 0.7 | Vegetation | 0.990 |

MODIS MOD11A1 V6.1 data were used to generate MODIS LST images in GEE. These data are ready-made data processed by USGS with the split-window algorithm [40] and converted to LST [32].

After the LST images were obtained from Landsat and MODIS, the statistical data of the images were evaluated, and the correlation between the mean values of the images was calculated. The observed differences are mainly due to the use of different retrieval algorithms used in the computation [29].

### 2.2.2. Statistical Analysis

The correlation coefficients were calculated using Pearson correlation method, which is a measure to calculate linear dependency between two variables, and its formula is given in Equation (6) [51]:

$$r_{xy} = \frac{\sum(x_i - \overline{x})\sum(y_i - \overline{y})}{\sqrt{\sum(x_i - \overline{x})^2}\sqrt{\sum(y_i - \overline{y})^2}} \tag{6}$$

where $\overline{x}$ is the average of $x$ variable, $\overline{y}$ is the average of $y$ variable, and $r_{xy}$ is the Pearson correlation coefficient of $x$ and $y$ variables and varies between $-1$ and $1$. If $r_{xy}$ is close to 1, the variables are positively correlated; otherwise, the variables are negatively correlated. After the correlation analysis, linear regression models were created using satellite and ground-based meteorological data. Linear regression analysis is based on the quantitative relationships between variables [52] and can be described as a linear equation like Equation (7):

$$Y = a_0 + a_1X_1 + a_2X_2 + \ldots + a_nX_n \tag{7}$$

where $X$ and $Y$ describe the variables, and $a_0$ to $a_n$ are the coefficients in the linear regression models.

The significance of the regression analysis results was evaluated with the $p$-value. The $p$-value is a primary value used to measure the statistical significance of the results of a hypothesis test, and a $p$-value less than 0.05 is considered statistically significant. $p$-values are typically determined using $p$-value tables or spreadsheets/statistical software. These calculations are based on assumptions or known probability distributions of the particular statistic being tested. The $p$-value is calculated from the deviation between the observed value and the chosen reference value, taking into account the statistical probability distribution; the greater the difference between the two values, the lower the $p$-value. Mathematically, the $p$-value is obtained by integrating the area under the probability distribution curve of all statistics that are, at least, as far from the reference value as the observed value for the total area under the probability distribution curve calculated [53]. In this study, $p$-values were calculated using Excel data analysis tools.

The results of the regression models were validated with meteorological station data for 2022 using root mean square error (RMSE) (Equation (8)):

$$RMSE = \sqrt{\frac{1}{N}\sum_{i=1}^{n}(Y_i - Yp_i)^2} \tag{8}$$

where $Y_i$ is the actual value, $Yp$ is the obtained value, and $n$ is the number of validation samples.

### 2.2.3. Thematic Mapping of Meteorological Parameters

For this purpose, correlation analysis was performed between the meteorological station data (for two meteorological parameters) between 1987 and 2017 and eight Landsat LST images produced on the same dates. The same procedures were applied to MODIS LST images, and correlation analyses were performed. Since the station data of the previous years are especially few, the data sets of all years were combined, and a correlation analysis was performed.

Due to the correlations obtained being satisfactory, the regression models were created using two meteorological parameters. Afterward, thematic maps were produced for two parameters using 2022 LST images, and their spatial distributions were mapped. For the accuracy assessment, RMSEs were taken into account, and the difference between the 2022 meteorological station data and the values obtained from the thematic map was calculated for each meteorological parameter considered.

## 3. Results and Discussion

Istanbul Landsat LST maps were created using nine cloud-free (1987–2022) Landsat images. However, since MODIS has provided data since 2000, six MODIS LST maps were created. The mean values of each LST image were determined, and the mean value differences calculated are given in Table 3.

**Table 3.** The mean values and differences calculated.

| Data (Mean, °C) | 26 July 2000 | 3 July 2003 | 19 July 2009 | 30 July 2013 | 9 July 2017 | 23 July 2022 |
|---|---|---|---|---|---|---|
| MODIS LST | 33.16 | 36.55 | 32.86 | 34.45 | 30.52 | 31.08 |
| Landsat LST | 31.52 | 34.28 | 29.04 | 32.4 | 29.04 | 30.47 |
| Difference, Δ | 1.64 | 2.27 | 3.82 | 2.05 | 1.48 | 0.61 |

In Table 3, it can be seen that there are 0.61 to 3.82 °C differences between Landsat LST and MODIS LST image mean values, while these differences are primarily related to the image acquisition time of the sensors (9:00 and 11:30 a.m. UTC, respectively). In addition, the fact that thermal sensors have different spectral and spatial resolutions can be considered as a secondary and third factor. The fourth factor is the use of two different algorithms for LST calculations from two different sensors.

The Pearson correlation coefficient was calculated between the Landsat LST and MODIS LST mean values given in Table 3, and it can be found that the MODIS LST and Landsat LST results are highly correlated with each other (Figure 4). As shown, a high correlation of 0.88 was found between Landsat LST and MODIS LST images. This result is consistent with a similar literature study conducted in Antalya, where the correlation was found to be 0.70 [16]. In another study conducted in China, for Landsat LST values and MODIS data, correlations were in a range of 0.85 [29].

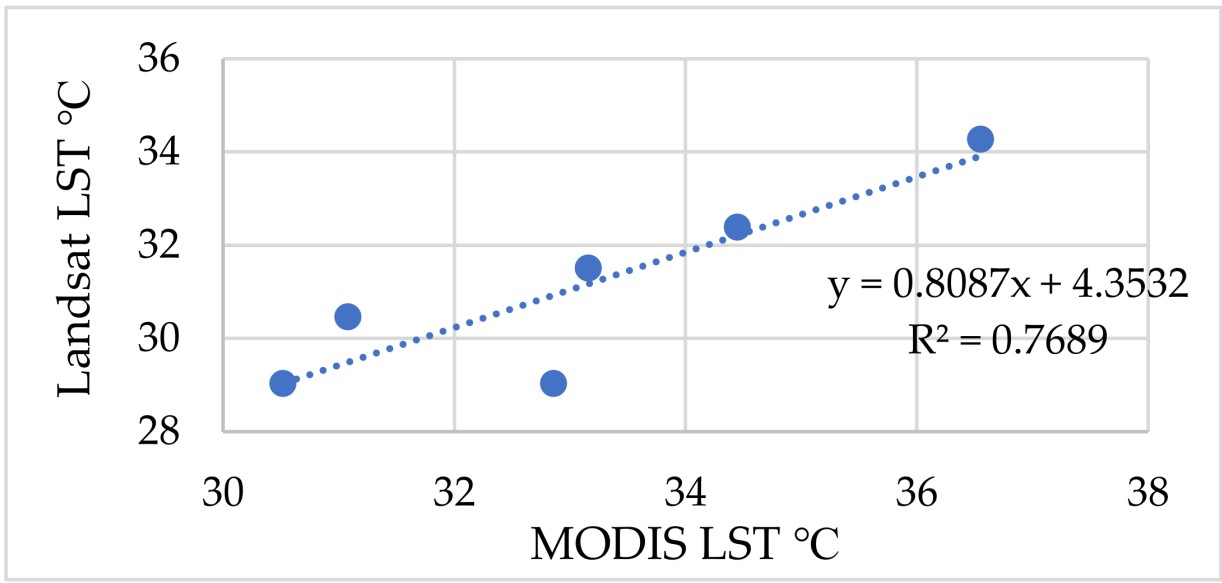

$y = 0.8087x + 4.3532$

$R^2 = 0.7689$

**Figure 4.** Landsat LST and MODIS LST images mean values and correlation.

Correlation analyses were performed between the station data of eight images (for each meteorological parameter) and the corresponding pixel values in the relevant Landsat LST images between 1987 and 2017. Since the Landsat 8 OLI/TIR and Terra MODIS's image acquisition times for Istanbul are approximately 09:00 UTC and 11:30 UTC, respectively, the meteorological data corresponding to these hours were used. The correlation coefficients calculated for each year and the number of station data used are given in Figure 5a,b.

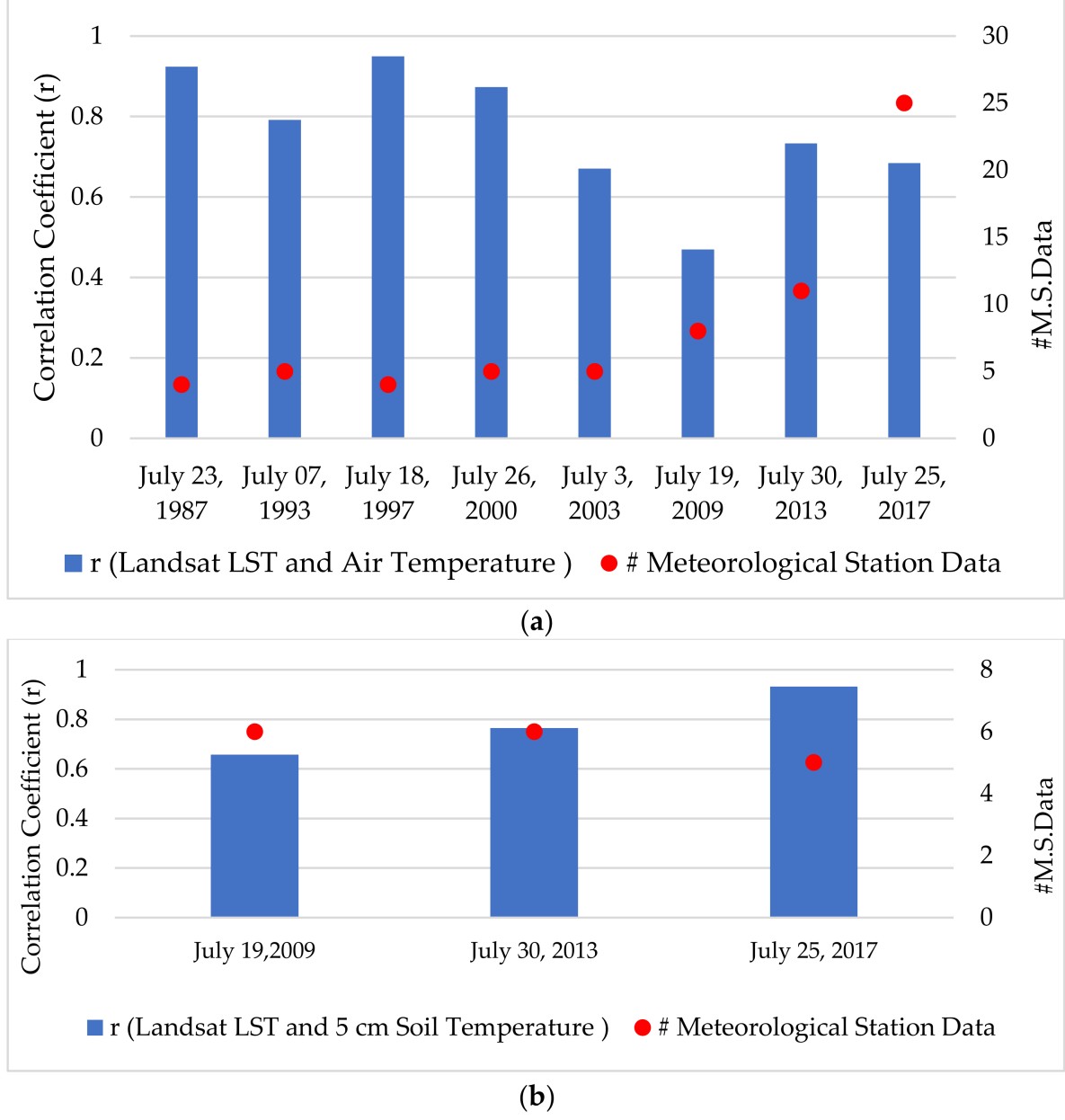

**Figure 5.** Correlation coefficients between Landsat LST and meteorological parameters ((**a**) air temperatures, (**b**) 5 cm soil temperatures) and number of data measured at the station.

When the correlations between the air temperature data from meteorological stations and Landsat LSTs for eight different dates in Figure 5a are examined, the lowest value is r = 0.47, and the highest is r = 0.95. When the correlations between the 5 cm soil temperature data from meteorological stations and Landsat LSTs for three different dates in Figure 5b are examined, the correlations are above 0.67. These results are consistent with the study [29],

which showed that the correlation of Landsat LST values with air temperature data was 0.82.

The correlation analysis results between the meteorological station data (for two meteorological parameters) and the corresponding pixel values in the relevant MODIS LST images between the years 2000 and 2017 are given in Figure 6a,b. According to the results, the correlation between the meteorological data and MODIS LST is high.

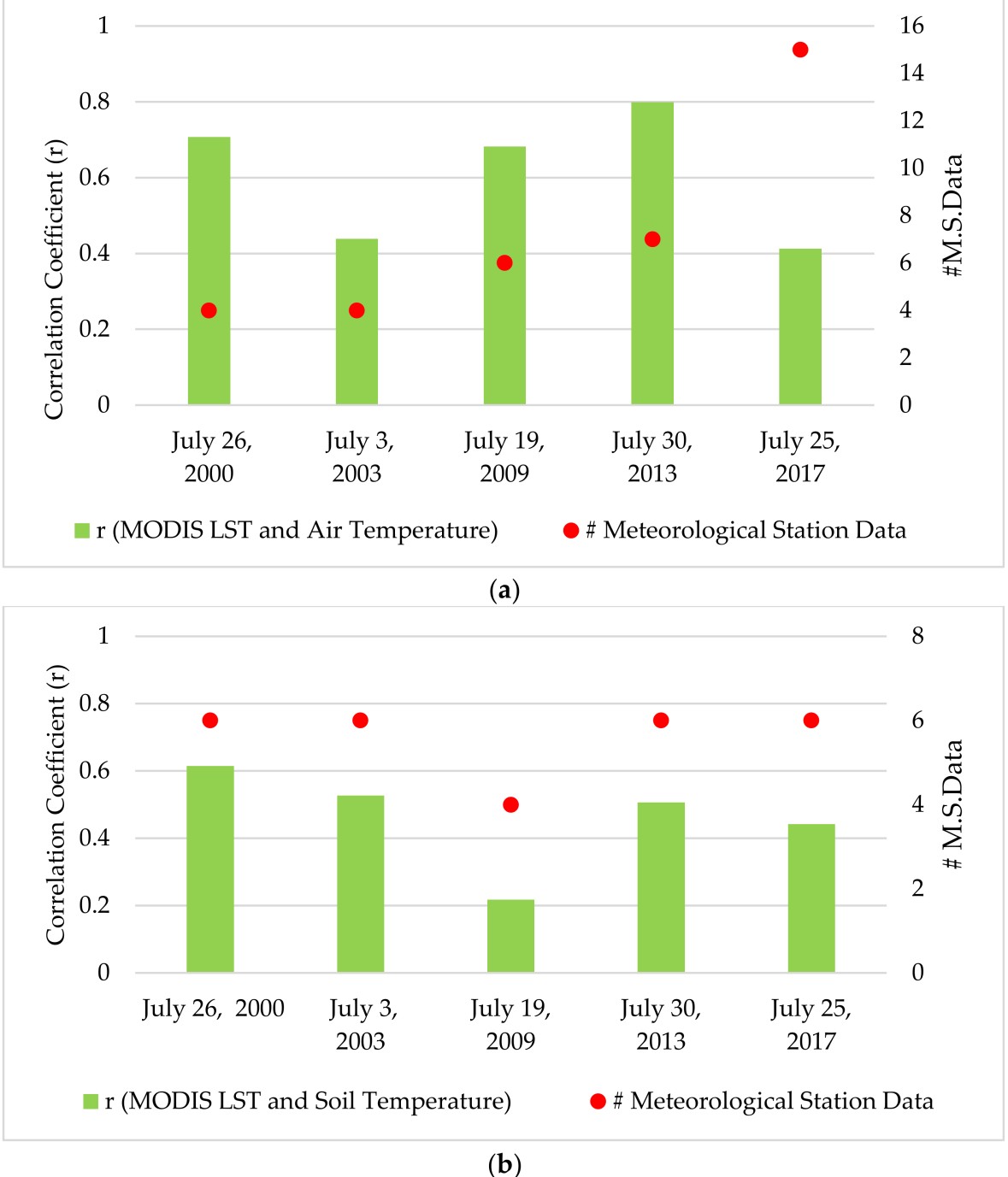

**Figure 6.** Correlation coefficients between MODIS LST and meteorological parameters ((**a**) air temperatures, (**b**) 5 cm soil temperatures) and number of data measured at the station.

When the correlations between the average temperature data from meteorological stations and the MODIS LSTs for five different dates in Figure 6a are examined, the lowest

value is r = 0.41 for 2017, and the highest value is r = 0.80 for 2013. When the correlations between the soil temperature data obtained from meteorological stations and MODIS LSTs for five different dates in Figure 6b are examined, the correlations are above 0.44, except for 19 July 2009 (r = 0.22). In the correlation analysis made with two different satellite data sets, it can be noticed that MODIS LST satellite data give a lower correlation result, which is thought to be mostly due to their lower spatial resolution.

As seen in Figures 5 and 6 above, there are data pairs with high correlation coefficients for each parameter and those with low correlation. Correlation analysis and regression analysis were performed using data from all years together, for each parameter, to see the effect of using data from all years. The correlations for individual years and all years are given in Table 4, together with the *p*-values obtained from the analyses.

**Table 4.** The correlations and *p*-values.

| Satellite Data | Meteorological Station Data | Year | Σ Point | r² | *p* |
|---|---|---|---|---|---|
| Landsat LST | Air Temperature | 1987 | 4 | 0.85 | 0.0761 |
| | | 1993 | 5 | 0.63 | 0.1107 |
| | | 1997 | 4 | 0.90 | **0.0500** |
| | | 2000 | 5 | 0.76 | 0.0533 |
| | | 2003 | 5 | 0.45 | 0.2158 |
| | | 2009 | 8 | 0.22 | 0.2403 |
| | | 2013 | 11 | 0.54 | **0.0103** |
| | | 2017 | 25 | 0.47 | **0.0001** |
| | | Total (8 years) | 68 | 0.53 | $\mathbf{2.52 \times 10^{-12}}$ |
| MODIS LST | Air Temperature | 2000 | 6 | 0.50 | 0.1159 |
| | | 2003 | 6 | 0.19 | 0.3837 |
| | | 2009 | 8 | 0.47 | 0.0625 |
| | | 2013 | 11 | 0.64 | 0.0032 |
| | | 2017 | 25 | 0.17 | 0.0363 |
| | | Total (5 years) | 56 | 0.26 | **0.00005** |
| Landsat LST | 5 cm Soil Temperature | 2009 | 5 | 0.87 | **0.0208** |
| | | 2013 | 6 | 0.58 | 0.0766 |
| | | 2017 | 6 | 0.58 | 0.0766 |
| | | Total (3 years) | 17 | 0.50 | **0.0016** |
| MODIS LST | 5 cm Soil Temperature | 2000 | 6 | 0.38 | 0.1944 |
| | | 2003 | 6 | 0.28 | 0.2824 |
| | | 2009 | 4 | 0.05 | 0.7828 |
| | | 2013 | 6 | 0.26 | 0.3051 |
| | | 2017 | 6 | 0.20 | 0.3800 |
| | | Total (5 years) | 28 | 0.44 | **0.00012** |

Statistically significant correlations are marked in bold in Table 4 ($p \leq 0.05$). When the *p*-values are examined in Table 4, it can be seen that for the Landsat LST and air temperature, *p* = 0.05 for 1997, *p* = 0.01 for 2013, and *p* = 0.0001 for 2017. The highest *p*-value was obtained using all data between 1987 and 2017, $p = 2.52 \times 10^{-12}$. Similarly, the best *p*-value between MODIS LST data and air temperatures was obtained using the total values, *p* = 0.00005. While the *p*-value between Landsat LST and soil temperature was 0.0208 for 2009, it was obtained as *p* = 0.0016 using the total data of 3 years. While it was observed that the *p*-values for each year were bigger than 0.05 between MODIS LST and soil temperatures, *p* = 0.00012 was obtained with all data of 5 years.

Thematic maps were produced (for data with $p \leq 0.05$) using 2022 Landsat LST and MODIS LST images and linear regression models, and the regression models used are given in Table 5. Accuracy assessments were made using 2022 meteorological station data for each thematic map produced, and RMSEs were calculated (Table 5).

When Table 5 is examined, the highest accuracy in the four air temperature maps produced with Landsat LST was obtained with the 2017 regression model and the regression

model using data from all years (RMSE 1.25 °C and 1.35 °C). In the three thematic maps produced with MODIS LST data, the lowest RMSE value (1.26 °C) was obtained with the regression model using the five-year data. Similarly, models created with total data in soil temperature thematic maps produced better results than models created from a single year (3.28 for Landsat, 9.05 for MODIS). In the accuracy assessment, the 2022 air temperature data from 31 stations were used for the validation of the 2022 air temperature maps derived; however, the soil temperature thematic maps derived could only be verified with a small number of measurements collected from seven stations. This clearly confirms the low soil temperature model accuracy in the analysis (Table 5).

**Table 5.** Linear regression models (with $p \leq 0.05$) and RMSE errors.

| Parameters | Data | Year | Linear Regression Models | RMSE (°C) |
|---|---|---|---|---|
| Air Temperature | Landsat LST 2022 | 1997 | $y = -0.7436x + 42.293 \; r^2 = 0.9024$ | 10.95 |
| | | 2013 | $y = 0.6355x + 9.5679 \; r^2 = 0.5373$ | 1.42 |
| | | 2017 | $y = 0.4099x + 16.685 \; r^2 = 0.4686$ | 1.25 |
| | | All (8 years) | $y = 0.4952x + 14.295 \; r^2 = 0.5268$ | 1.35 |
| Air Temperature | MODIS LST 2022 | 2013 | $y = 0.6725x + 9.0265 \; r^2 = 0.6383$ | 8.54 |
| | | 2017 | $y = 0.2468x + 22.842 \; r^2 = 0.1602$ | 1.35 |
| | | All (5 years) | $y = 0.3443x + 19.914 \; r^2 = 0.2585$ | 1.26 |
| 5 cm Soil Temperature | Landsat LST 2022 | 2009 | $y = 3.8288x - 83.763 \; r^2 = 0.8696$ | 9.32 |
| | | All (3 years) | $y = 0.839x + 3.7863 \; r^2 = 0.4977$ | 3.28 |
| 5 cm Soil Temperature | MODIS LST 2022 | All (5 years) | $y = 1.7682x - 19.925 \; r^2 = 0.4395$ | 9.05 |

The air temperature map and 5 cm soil temperature map derived using the 2022 Landsat LST map and regression models (generated with all-year data) are given in Figure 7. In the accuracy evaluation of thematic maps, data from 31 stations were used as 2022 air temperature data. Since the soil temperature was measured at seven stations, it could be evaluated with a small number of measurements.

The air temperature map and 5 cm soil temperature map derived using the 2022 MODIS LST map and regression models (generated with all-year data) are given in Figure 8.

When Figures 7 and 8 above are examined, it can be seen that Landsat LST and MODIS LST maps are similar, and more detailed maps are obtained with Landsat, which has a much higher spatial resolution. It is thought that one of the reasons for the high-temperature values in the MODIS LST maps is related to the time the image was taken. Specifically, the Landsat 8 OLI/TIR acquisition time for Istanbul is approximately 09:00 UTC, while Terra MODIS's is approximately 11:30 UTC. In the LST maps, the air temperatures are higher in the urban areas, while the temperatures are lower in the forest areas located to the north of the city. The same situation can be observed in the air temperature and soil temperature maps.

While air temperature maps derived from Landsat and MODIS LST maps have high RMSE accuracies, temperatures are also in similar ranges. When the soil temperature maps derived from Landsat and MODIS LST maps are examined, the Landsat soil temperature map (RMSE = 3.28 °C) is in the range of 22–38 °C, while the range in the MODIS soil temperature map (RMSE = 9.05 °C) is 25–45 °C. Therefore, it can be said that Landsat LST gives better results in deriving soil temperature maps according to RMSE values. In addition, the differences in the orbital periods of the satellites also cause a difference between the sensed temperature values, which affects the accuracy of the regression models. As a result, there will be a slight difference between the spatial distribution maps derived by these two satellites.

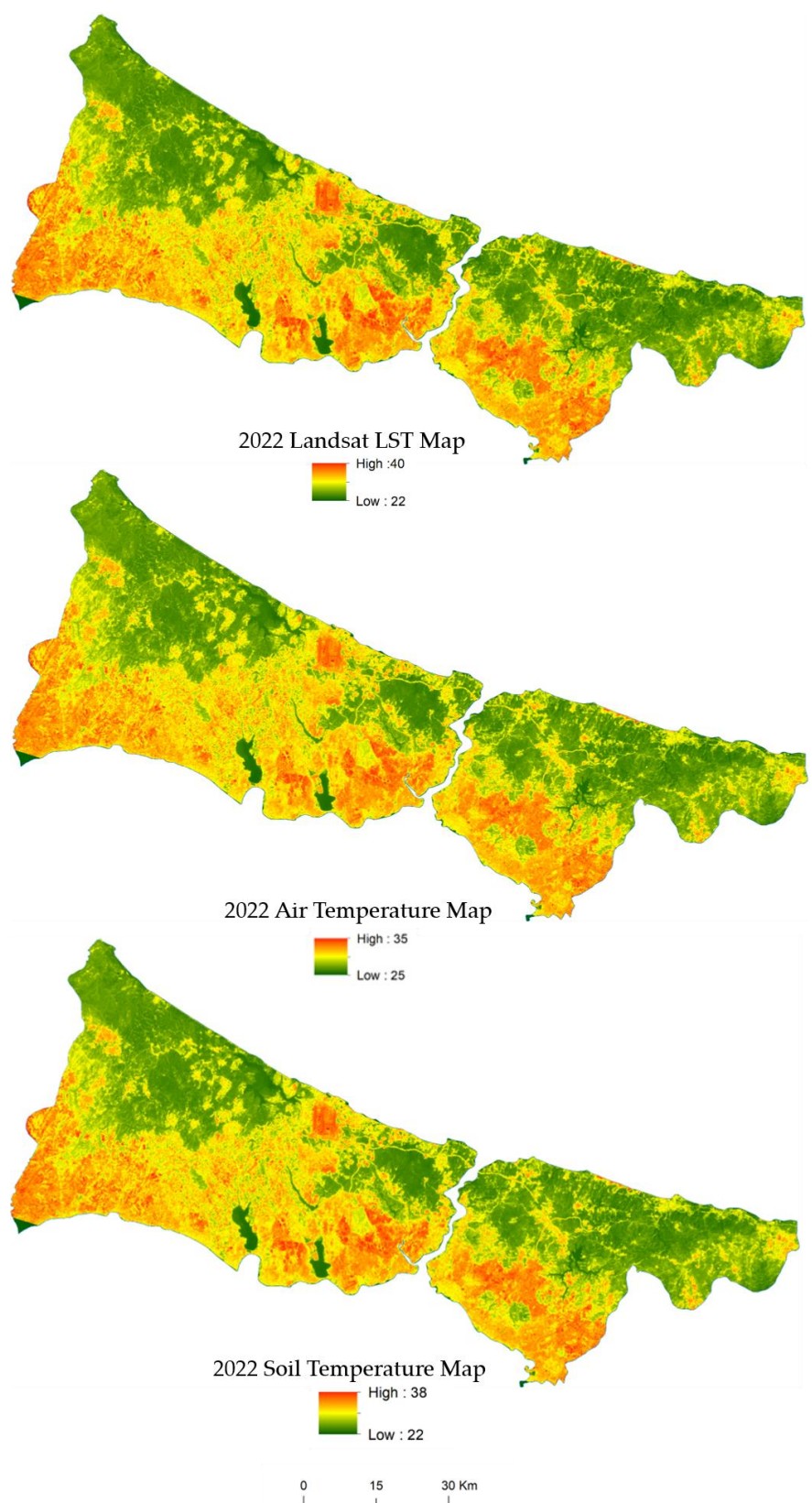

**Figure 7.** Landsat LST (°C) map dated on 23 July 2022 and thematic maps (air temperature and soil temperature) derived from regression models (for 09:00 a.m. UTC).

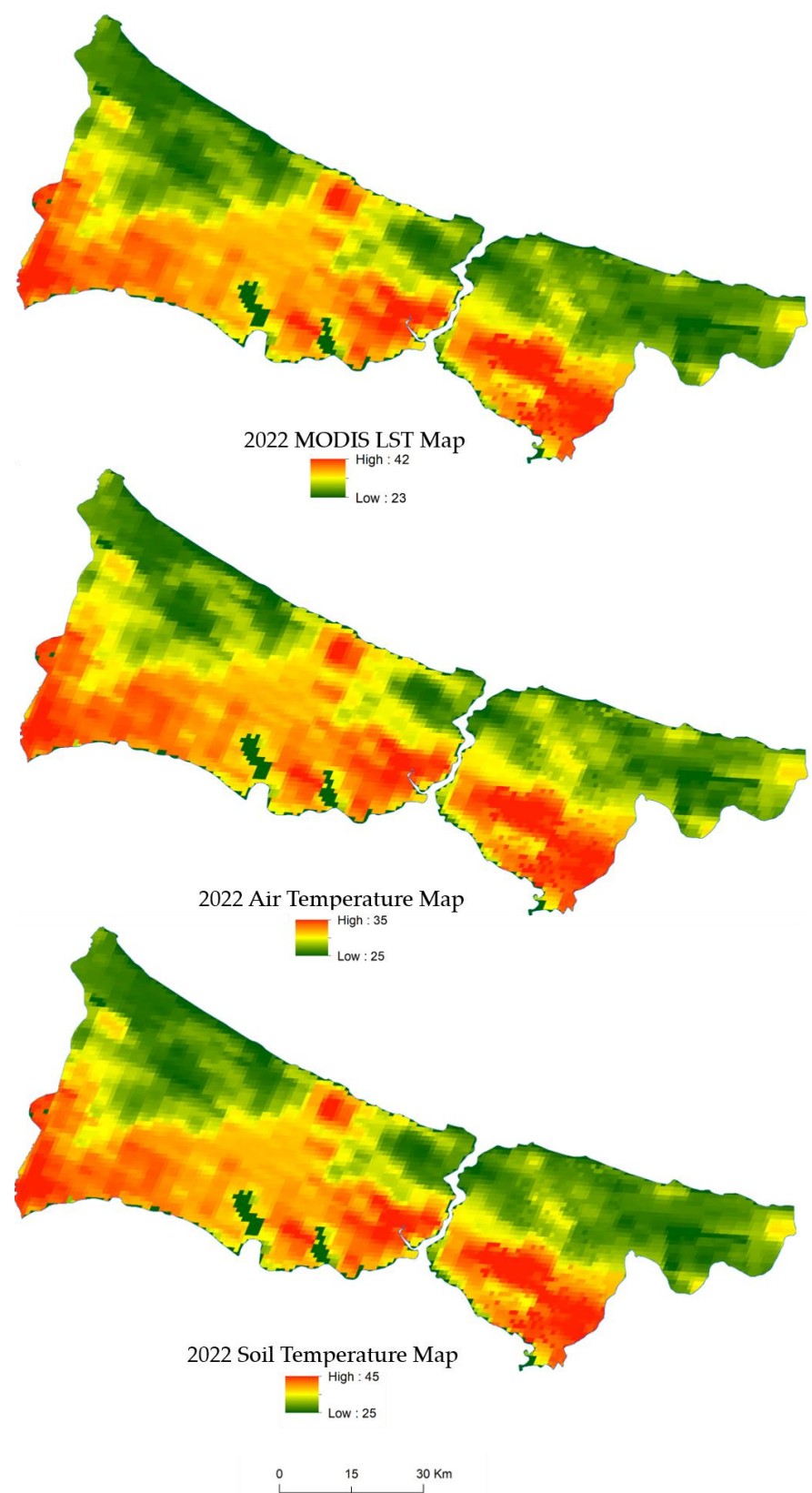

**Figure 8.** MODIS LST (°C) map dated on 23 July 2022 and thematic maps (air temperature and soil temperature) derived from regression models (for 11.30 a.m. UTC).

## 4. Conclusions

With its spatial and temporal resolution capabilities and continuous data availability, remote sensing technology and data offer a great opportunity to monitor environmental impacts in megacities. In this study, the linear correlations between the meteorological station data (i.e., air temperature and 5 cm soil temperature) and satellite-derived LSTs in the heterogeneous megacity Istanbul were investigated. Although Istanbul is the most significant city in Turkey, only a small number of the 54 available meteorological stations had data on hourly air temperature and hourly soil temperature. Therefore, the applicability of satellite-based LST data for the derivation of meteorological parameters was examined, in this study, in the absence of adequate hourly data.

In order to derive the spatial distribution maps of the two meteorological parameters, linear regression models were first created between the LST data and the meteorological station data from previous years due to the sufficient correlations found. The accepted regression models were then applied to the 2022 LST data.

The accuracies of the obtained thematic maps were examined using 2022 meteorological station data. In general, it was seen that the accuracy of thematic maps derived with the regression models created using data from all years is higher than the maps derived with each year's data. The RMSE values for the air temperature map produced by Landsat LST and MODIS LST were obtained as 1.35 °C and 1.26 °C, respectively. According to the results of the accuracy analysis performed with a small number of station data, it was observed that the soil temperature maps had lower RMSEs (i.e., RMSE = 3.28 °C with Landsat LST, RMSE = 9.05 °C with MODIS LST).

The main findings obtained are (i) the meteorological data derived using satellite-based LSTs can be used as in situ data in cases where meteorological station data are not available and/or insufficient. Accuracy analyses, in particular of air temperature thematic maps derived by regression models using 2022 Landsat LST and MODIS LST data, provide evidence of this; (ii) despite the low resolution of MODIS LST data, which provides daily data, its performance was also remarkable; and (iii) in addition, a high correlation was determined between the mean values of the Istanbul MODIS LST maps and Istanbul Landsat LST maps (r = 0.88).

As a future study, the correlation of LSTs with different meteorological data will be evaluated on the basis of land use/land cover types in not only heterogeneous but also homogeneous urban areas. In addition, an analysis will be carried out considering other seasons.

**Funding:** This research received no external funding.

**Institutional Review Board Statement:** Not applicable.

**Informed Consent Statement:** Not applicable.

**Data Availability Statement:** Not applicable.

**Acknowledgments:** I would like to thank the Turkish General Directorate of Meteorology for providing free data.

**Conflicts of Interest:** The author declares no conflict of interest.

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
