# Peer review of "Investigation of the Efficiency of Satellite-Derived LST Data for Mapping the Meteorological Parameters in Istanbul"

_atmosphere, doi:10.3390/atmos14040644_

Round 1
Reviewer 1 Report (Previous Reviewer 1)
In my opinion, the manuscript has undergone significant improvement since its initial submission and the revisions made in response to previous reviews have addressed many of the concerns raised. The language used in the paper is now clear and concise, making the content easy to understand and follow. However, there are still a few minor issues that need to be addressed to ensure the manuscript meets the standards of the journal. With these final revisions, the manuscript will be ready for publication and will make a valuable contribution to the field.
1. Line 26. I do not consider 'Linear Correlation' to be an appropriate keyword.
2. Line 27. Change 'Soil temperature' to 'Soil Temperature'.
3. Table. 5. I recommend that the 'r' column be removed as it does not serve a meaningful purpose in this context.
Author Response
Thank you for all your valuable contributions. Kind regards.

Reviewer 2 Report (Previous Reviewer 2)
After revision, the manuscript improves a lot. All the comments are justified or added. It can be accepted in its present form.
Author Response
Thank you for all your valuable contributions. Kind regards.
Reviewer 3 Report (Previous Reviewer 3)
In general, I rate the originality and scientific level of the reviewed article rather low, but after the changes made by the author it can be published. I still found a few minor editorial errors and factual errors in the text (marked next to the text in the attached PDF file). Please correct them.

Author Response
Dear reviewer,
I made the corrections you marked. Thank you very much for all your attention and comments. You have contributed a lot to this article.
Thank you very much for your valuable contribution.
Adalet

This manuscript is a resubmission of an earlier submission. The following is a list of the peer review reports and author responses from that submission.
Round 1
Reviewer 1 Report
The authors map four meteorological parameters using satellite-derived LST data. This study can provide new insights for real-time weather station parameter mapping. The topic is interesting and I enjoyed reading it in detail. Nevertheless, I have some concerns on the methodology/quality of the manuscript (see comments below) and, in my opinion, the manuscript needs significant improvements before it can be considered for publication in Atmosphere.
Major Concerns:
1. The manuscript structure/content should be improved. Specifically:
1.1. The Introduction does not provide a sufficient overview of the existing literature on the correlation between satellite-derived LST and meteorological parameters. In addition, the novelty of the paper is not demonstrated by a deep analysis of the state-of-the-art; in other words, I am aware about many other studies on the same topic that have not been considered by the authors. I think the manuscript would benefit from a brief (but detailed) discussion on key processes and main results in the literature.
1.2. The results of the "method" in the "Results" section should be mentioned in the "Methods" section and vice versa. For example, correlation analysis (Pearson or Spearman or other correlation analysis?) need to be specified in the "Methods" section; regression models? Which models? Linear? Power? Or ? Please specify in the Method and Results section. I suggest the authors add the "Statistical analysis" subsection in the M&M section.
1.3. The manuscript has no Conclusions – it ends with a “Discussion” section and the reader is left wondering what the take-home message is.
1.4 Some content can be combined to improve readability. For example, Table 1 and Table 2 can be combined into one Table. Also, Table 6 and Table 8 can be combined into one Table…
2. Many paragraphs in the Results section have only one sentence, which need to be readjusted. For example, Lines 188-189; Line 234.
3. The Discussions section is too succinct and confusing, some comparisons with literature data are incomplete or only outlined, without entering deeply in them. Therefore, I suggest to thoroughly rewrite this section, focusing the main concepts.
Minor Concerns:
1. Page 1, Line 11. Please change ‘land surface temperature’ to ‘LST’.
2. Page 6, Line 179. Please list the formula of RMSE or cite the corresponding literature.
3. Page 9, Lines 220-225. The font size of this paragraph looks different from others. Please check it.
4. Page 11 Line 245. Please add the unit of RMSE.
Author Response
Dear Reviewer,
The corrections requested by you have been made, and the answer to your requests is below.
Thank you very much for all your valuable comments and contributions to the article.
Kind regards.

Reviewer 2 Report
The manuscript titled “Investigation of the Efficiency of Satellite-Derived LST data for Mapping the Meteorological Parameters in Istanbul” shows promising results. The study has good potential. Methodology and datasets are acceptable. However, some following issues should be modified:
· Is there any novelty found in the methodology of this study?
· The manuscript needs a comparison with the recently published articles. There are many recent similar research articles available on some global cities. Kindly consider these articles.
Author Response

(The authors gave the same response as above.)

Reviewer 3 Report
Wyniki tłumaczenia
TÅ‚umaczenie
The article deals with important issues and one of the largest urban agglomerations in the world, but unfortunately the level of elaboration is very low. In my opinion, there is not much chance to correct errors and fill gaps. The article should be reworked from the very beginning. Detailed comments have been placed with the text of the article in the attached PDF file. I will mention here only the most important ones concerning the applied methodology. In my opinion, the methodology used is wrong and therefore the results obtained are of little use. 1. LST satellite images represent instantaneous values as they are scanned by the satellite sensor. Therefore, for each image used, the exact hour and minute of its capture should be provided. 2. What is the point of studying the relationship between the instantaneous LST value from the satellite image and the average daily values of several meteorological parameters? In my opinion, the measurement values of the meteorological parameters from the timely measurements closest to the time of the satellite scan should be used. 3. Apart from the information on the use of cloud-free satellite images taken in July, there are no other data important for the interpretation of the results: what were the circulation conditions (inflow of air masses) in the period before and during the acquisition of satellite images, and more generally - how the conditions during the image taking were compared to the average (extreme) meteorological conditions in Istanbul in July. This is the only way to assess the representativeness of the results and the possibility of drawing more general conclusions from them. Assuming that it takes 1 minute for the satellite to scan the Istanbul area, we can calculate the proportion of the sample to the total population. 36 July months of 31 days are 1116 days. Each day is 1440 minutes. This results in a population of 1,607,040 minutes for Landsat images. The 8 images used are 0.000004978 in total (nearly five millionths). In addition, this is not a random sample, but a preferential one, because it does not cover cloudy periods. In this context, it would be useful to know what is the average cloud cover in Istanbul in July during the whole day (24 hours) and during the day when the sun is shining. 4. Within 36 years (1987-2022), significant changes have certainly occurred in the study area, both climatic and related to land cover and land use, especially the share of built-up area. It certainly had a significant impact on the spatial distribution of LST and the relationship between LST and meteorological parameters. The characteristics of these changes should be an important element of the description of the research area. Unfortunately, she's not there. 5. What was the land cover and use within the Landsat (TM/TIRS) and MODIS pixels where the weather stations were located? Has it changed significantly (particularly large MODS pixels) over the last 22 years? 6. Considering that the temporal representativeness of the analyzed sample (the dates from which the satellite images were derived) is unknown, all this analysis of temporal trends is meaningless (unreliable).
Author Response
Dear reviewer, thank you for your suggestions and careful corrections. I think it's a much better article now; I hope you will agree.
Kind regards
Adalet

Round 2
Reviewer 1 Report
There are only two minor issues here
1. In Table 3, please change ‘0.1≤ NDVI ≤ 07’ to ‘0.1≤ NDVI ≤ 0.7’;
2. Lines 337-338. Please add the unit of RMSE.
3. Figure 7 and 8. Please add the unit of LST.
Reviewer 3 Report
Although the author of the reviewed article made significant changes and additions to the first version, it did not change my opinion. The scientific level of this study is still too low to be published in the journal Atmosphere. There is nothing in this study, apart from a specific area of research, that could be considered a new scientific achievement. The interpretation of the results is very limited and superficial. Detailed comments have been included in the text of the article in the attached PDF file.
